Comprehensive analysis of DNA methylation patterns in recurrent miscarriage: imprinted/non-imprinted genes and their regulation across sperm and fetal-maternal tissues

Niu Yanru 1
Yin Lanlan 2
Zhou Yulan 2
Pang Xiaoyan 2
Li Yunqinq 2
Peng Cailing 2
Yao Meihua 2
Zhang Guoling 2
Yang Kaijie 2
Ma Tianzhong mtz@gdmu.edu.cn 2
1 Orthopaedic Research Lab, Affiliated Hospital of Guangdong Medical University , Zhanjiang , Guangdong Province , China
2 Reproductive Medicine Center, Affiliated Hospital of Guangdong Medical University , Zhanjiang , Guangdong Province , China
Nunes-da-Fonseca Rodrigo
Electronic publication date: 2025 Oct 7
Publication date: 2025
Volume: 13
Electronic Location ID: e20125
Received 2025 Feb 20; Accepted 2025 Sep 1
Copyright: ©2025 Niu et al.
Copyright year: 2025
Copyright holder: Niu et al.
License: This is an open access article distributed under the terms of the Creative Commons Attribution License, which permits unrestricted use, distribution, reproduction and adaptation in any medium and for any purpose provided that it is properly attributed. For attribution, the original author(s), title, publication source (PeerJ) and either DOI or URL of the article must be cited.
License URL: https://creativecommons.org/licenses/by/4.0/

Keywords: Recurrent miscarriage, DNA methylation, Epigenetics, Imprinted genes, Chorionic villi, Decidua, Sperm, Differentially methylated probes (DMPs)

Funding: National Nature Science Foundation of China 81300484 Research Initiation Program for High-level Talents of the Affiliated Hospital of Guangdong Medical University GCC2023016 Zhanjiang Science and Technology Special Fund Project 2022A01169 2024B01241 Scientific Research Fund Project of Guangdong Medical University M2017014 This study was supported by funding from the National Nature Science Foundation of China (81300484), Research Initiation Program for High-level Talents of the Affiliated Hospital of Guangdong Medical University (GCC2023016), Zhanjiang Science and Technology Special Fund Project (2022A01169, 2024B01241), Scientific Research Fund Project of Guangdong Medical University (M2017014). The funders had no role in study design, data collection and analysis, decision to publish, or preparation of the manuscript.

==============================
Background

Epigenetic regulation, including DNA methylation, is essential for normal embryonic development and maternal-fetal interactions. Recurrent miscarriage (RM), defined as the loss of two or more consecutive pregnancies, poses significant clinical and emotional challenges. However, the role of epigenetic alterations in RM, particularly in gametes and placental tissues, remains underexplored. This study aims to investigate global DNA methylation profiles and imprinting gene expression in the context of RM, providing insights into epigenetic mechanisms that contribute to pregnancy failure.

Methods

Genome-wide DNA methylation profiling was performed on sperm and chorionic villi from RM patients and control couples undergoing artificial abortion using the Illumina HumanMethylation 450 BeadChip platform. Genes related to differentially methylated probes (DMPs) in functionally critical genomic regions, including enhancers, promoters, and DNase hypersensitive sites (DHS), were identified and submitted to Gene Ontology (GO) and Kyoto Encyclopedia of Genes and Genomes (KEGG) enrichment analyses. Key imprinting genes (CPA4 and PRDM16) were validated at the protein level using Western blotting.

Results

RM samples exhibited a significant increase in hypermethylated DMPs across all analyzed tissues, with villi and decidua showing the highest numbers of epigenetic changes. Enrichment analyses highlighted pathways implicated in tissue morphogenesis, immune regulation, and cell signaling, including the PI3K-Akt, TGF-beta, and Wnt signaling pathways. Among imprinting genes, CPA4 and PRDM16 showed distinct hypomethylation at enhancer regions, corresponding to elevated protein expression in RM villi tissues.

Conclusions

This study identifies profound epigenetic dysregulation in RM-associated tissues, emphasizing the contribution of imprinting gene methylation abnormalities to pregnancy loss. Future studies incorporating functional assays and animal models are essential to elucidate the causal roles of candidate genes in RM pathogenesis and maternal-fetal health.

Introduction

Recurrent miscarriage (RM) refers to the loss of two or more consecutive pregnancies prior to the 20th week of gestation. Among couples attempting to conceive, its incidence is approximately 1%–3% (Tsonis et al., 2021). Recurrent miscarriage not only poses physical and mental challenges to patients but also represents a thorny clinical problem with unclear etiologies. The causes of recurrent miscarriage are complex and diverse, involving multiple factors such as genetic and immune aspects (Arias-Sosa et al., 2018; Cavalcante, Sarno & Barini, 2021; Yue et al., 2016). However, in many cases, the specific causes of recurrent miscarriage remain unknown (Dai et al., 2023).

Epigenetics refers to heritable changes in gene expression, such as DNA methylation and histone modifications, which do not involve alterations in the underlying DNA sequence. It plays a crucial role in the process of embryonic development by regulating gene expression to guide and coordinate normal developmental processes (Bale, 2015; Quan et al., 2021). Moreover, factors like maternal nutritional status and environmental exposures can influence fetal gene expression through epigenetic mechanisms, thereby exerting long-term impacts on fetal development (Lan et al., 2013). Such impacts emphasize the significance of epigenetics in understanding the mechanisms of recurrent miscarriage.

Studies have demonstrated that abnormal DNA methylation is closely associated with RM (Demond et al., 2019). The changes in methylation levels in the chorionic villi and decidua tissues of patients with recurrent miscarriage may affect immune regulation, thereby influencing pregnancy outcomes (Wang et al., 2022). Sperm provides half of the genetic information for offspring, and the abnormal development of the embryo followed by miscarriage may also be influenced by the genetic factors of male sperm (Dada et al., 2012). The abnormal methylation levels at specific gene loci in sperm are related to unexplained recurrent miscarriage (Ankolkar et al., 2012). Most previous studies were limited to the comparison of one or two sample types, lacking a systematic analysis of the differences in DNA methylation among different sample types such as sperm, chorionic villi, and decidua (Ma et al., 2023; Matsumoto et al., 2022; Pi et al., 2020). Consequently, this has restricted the multi-dimensional understanding of recurrent miscarriage, especially the in-depth exploration of the synergistic effects of paternal and maternal factors in embryonic development and pregnancy maintenance as well as their underlying mechanisms.

The role of the epigenetic mechanism of imprinted genes in recurrent miscarriage is an important research area. Imprinted genes are a class of genes regulated by epigenetic mechanisms, and their expression is influenced by the parental origin, which plays a crucial role in fetal development (Tucci et al., 2019). Studies have shown that the abnormal expression of imprinted genes is closely related to placental dysfunction, which is an important factor contributing to intrauterine growth restriction (IUGR) and recurrent miscarriage (Canicais et al., 2021). It has also been found that the DNA methylation levels of specific imprinted genes are abnormal in patients with recurrent miscarriage, indicating that the epigenetic alterations of genes may be one of the potential causes of recurrent miscarriage (Liu et al., 2018). Moreover, in some studies, recurrent miscarriage is associated with abnormal methylation of imprinted genes in sperm (Cannarella et al., 2021). The connection between the abnormal methylation of imprinted genes and recurrent miscarriage has not been fully verified. Therefore, it is of great significance to comprehensively analyze the role of imprinted genes in the occurrence process of recurrent miscarriage by combining the epigenetic characteristics of multiple sample types.

This study aims to conduct a comprehensive analysis of the DNA methylation profiles in the fetal-maternal tissues and sperm of couples with recurrent miscarriage as well as those of control couples. By integrating the methylation data from different tissues and identifying differentially methylated probes (DMPs), we have identified the imprinted/non-imprinted genes associated with recurrent miscarriage. Furthermore, experimental verification of the key genes, such as Carboxypeptidase A4 (CPA4) and PR domain-containing 16 (PRDM16), helps to unravel the functional impacts caused by methylation alterations at these loci. These research findings will help to deepen our understanding of the overall epigenetic landscape related to recurrent miscarriage, and highlight the potential molecular markers and research pathways for future studies.

Materials & Methods

Sample collection and preparation

This study was approved by the Ethics Committee of the Affiliated Hospital of Guangdong Medical College (YJY2018084), and all subjects signed the informed consent forms. The inclusion criteria for patients with recurrent miscarriage were as follows: those who had experienced three or more miscarriages without known abnormal factors such as anatomical, chromosomal, endocrine, and immune disorders. A total of six samples from patients with unexplained recurrent miscarriage were collected, including three chorionic villi samples and three semen samples from each couple. Three chorionic villi samples and three semen samples of the normal control group were voluntarily provided by healthy couples who had undergone artificial accidental termination of pregnancy (AA). Using a stereoscopic microscope, the chorionic villi tissues were immediately separated from the samples after miscarriage or abortion. To eliminate somatic cell contamination, semen samples were processed by swim-up separation: motile sperm were isolated via migration into pre-equilibrated medium after 45–60 min incubation (37 °C/5% CO2), with subsequent phase-contrast microscopy (20 ×) confirmation of pure sperm pellets (>99% purity, no detectable somatic cells; Fig. S1). All the samples were then stored in a −80 °C freezer for subsequent DNA methylation detection.

DNA methylation detection

Frozen samples of the sperm and chorionic villi from three pairs of couples (participants) were thawed on ice. The tissues were minced, and then 180 µl of Buffer ATL and 20 µl of proteinase K were added, followed by incubation in a water bath at 56 °C for lysis. For semen samples, after centrifugation at 3,000× g for 10 min, the precipitate was resuspended in 200 µl of PBS. Subsequently, 20 µl of proteinase K was added and dithiothreitol was added to a final concentration of 0.04 mol/L, followed by incubation at 56 °C for 2 h (Chotiwan et al., 2017; Wimbles, Melling & Shaw, 2016). Upon completion of lysis, genomic DNA was extracted using the QIAwave DNA Blood & Tissue Kit (Qiagen, Cat. No. 69506, Batch No. 148047989) and purified using the QIAamp spin column according to the manufacturer’s instructions for later use.

The quality of DNA was quantified and detected by Nanodrop and agarose gel electrophoresis, and then subjected to bisulfite conversion treatment using the EZ DNA Methylation-Gold™ Kit (ZYMO RESEARCH, Cat. D5006). Following successful conversion, microarray-based DNA methylation detection was performed with the Infinium Human Methylation 450 BeadChip (Illumina) according to the manufactor’s instructions (Sandoval et al., 2011).

Public dataset integration

DNA methylation data from GSE141298 (six decidua samples of RM group, gestational age: 8.27 ± 1.87 weeks) (Pi et al., 2020) and GSE198700 (20 samples: five replicates each for RM/AA decidua and chorionic villi) (Matsumoto et al., 2022) were integrated. Both datasets were detected by Illumina Human Methylation 450 BeadChip. The gestational ages in GSE141298 matched our cohort’s range (Table 1), while GSE198700 showed no significant difference between RM (7.2 ± 0.8 weeks) and AA (7.1 ± 0.8 weeks; Mann–Whitney’s U test, P = 0.80).

Table 1 Statistics of clinical information for RM and AA patients.

	RM patients (n = 3)	AA controls (n = 3)	P value	
Male age	31.33 ± 4.04	32.33 ± 3.06	0.90	
Female age	30.33 ± 3.51	30.67 ± 2.52	0.75	
Gestational weeks	8.00 ± 1.00	8.33 ± 0.58	0.65	
Miscarriages	3.67 (3–5)	0 (0–0)	0.03	
Previous live births	0 (0)	1 (0–2)	0.23	
Notes.

Age was presented as mean ± standard deviation (SD). Pregnancies/Miscarriages and previous live births were presented as average (range).

RM recurrent miscarriage

AA artificial abortion

DNA methylation analysis

ChAMP v2.36.0 was utilized to conduct merging, quality control, normalization, and batch effect removal analyses on the DNA methylation chip data of the above 38 samples (Morris et al., 2014). Subsequently, the identification of DMPs between the RM and AA groups was carried out for chorionic villi, decidua tissues, and sperm samples, respectively. The probes with a P <  0.01 and an absolute value of Δβ >  0.1 were defined as DMPs. DMPs in DNA hypersensitive sites (DHS), enhancer, intergenic, and promoter regions were screened for subsequent functional enrichment analyses. The region within 1,500 base pairs around the transcription start site was defined as the promoters.

The annotations of human imprinted genes were obtained from the Geneimprint database (http://www.geneimprint.com/, accessed in June 2024). Genes marked as “Not Imprinted” and “Conflicting Data” were filtered out, and the remaining genes were retained as human imprinted genes. The genomic coordinates of the imprinting control regions (ICRs) were obtained from the Geneimprint database (Jima et al., 2022), and the DMPs in intergenic regions located in ICRs were identified.

Real-time quantitative PCR (qPCR)

Total RNA was extracted from homogenized chorionic villi samples using TRIzol reagent (Invitrogen, USA). Genomic DNA was eliminated, and first-strand cDNA synthesized from 1 µg RNA using HiScript III RT SuperMix for qPCR (+gDNA wiper) (Vazyme Biotech, China). qPCR amplification was performed on an ABI 7500 system (Applied Biosystems, USA) with ChamQ Universal SYBR qPCR Master Mix (Vazyme). Reactions (20 µl volume) included: initial denaturation at 95 °C for 30 s; 40 cycles of 95 °C for 10 s and 60 °C for 30 s. All samples were run in triplicate with no-template controls. β-actin served as the endogenous control. Relative expression was calculated by the 2−ΔΔCt method. Primer sequences are provided in Table S1.

Western blotting

The freshly collected chorionic villi tissue samples were placed into pre-cooled phosphate-buffered saline (PBS) and gently rinsed to remove impurities such as blood and mucus. Subsequently, the tissues were cut into pieces and then homogenized using a tissue homogenizer. Further, the samples were lysed with RIPA cell lysis buffer containing protease inhibitors for 30 min and centrifuged at 12,000× g for 15 min at 4 °C. The resulting supernatant was used for Western blotting analysis.

After electrophoresis, the protein samples were transferred onto polyvinylidene fluoride (PVDF) membranes, which were then blocked with 5% skim milk powder at room temperature for 1 h. CPA4 antibody (26824-1-AP, Proteintech) or PRDM16 antibody (55361-1-AP, Proteintech) was added respectively, and the membranes were incubated overnight at 4 °C. After being washed with Tris-buffered saline containing Tween (TBST), the membranes were incubated with secondary antibodies at room temperature for 2 h. Subsequently, the membranes were washed with TBST for three times. The bands were detected using a chemiluminescence (ECL) kit, and β-actin was used as a control. Three biological replicates were set for the RM and AA groups, respectively.

Statistical analyses

The inter-group difference analysis of clinical data and the quantitative results of Western blotting for imprinted genes was conducted using the Student’s t-test. A P <  0.05 indicates a statistically significant difference.

Results

Global DNA methylation profiles in gametes and fetal-maternal tissues

Sperm and chorionic villi from couples with RM and those from control couples with AA were selected for the detection of genome-wide DNA methylation profiles, each with three replicates, resulting a total of 12 samples. There were no significant differences in age between RM and AA groups for both male and female patients (Table 1; P = 0.90 and P = 0.75). The patients with RM had experienced an average of 3.67 miscarriages (ranging from 3 to 5 times), and none of the patients with recurrent miscarriage had given birth after multiple miscarriages (Table 1).

The DNA methylation data of the above 12 samples, six samples from GSE141298 (six decidua tissue samples of the RM group), and 20 samples from GSE198700 (five replicates each for chorionic villi and decidua tissues of the RM and AA groups) were integrated (Table S2). Among them, the gestational weeks of decidua samples in dataset GSE141298 were 8.27 ± 1.87 weeks, which is close to the dataset in this study (Table 1). The mean gestational weeks showed no significant differences between RM and AA for GSE198700 (7.2 ± 0.8 vs. 7.1 ± 0.8 weeks, P = 0.80). All of the above data were detected based on the Illumina Infinium HumanMethylation 450 platform. Through quality control, 420,765 CpG probes were filtered out from the original 485,577 probes for the next step of analysis (Fig. 1A). The results of cluster analysis showed that different sample types were located in different branches, indicating significant differences among tissue types (Fig. 1B). In chorionic villi tissues, decidua tissues, and sperm, 3,523 (Table S3), 2,415 (Table S4), and 253 DMPs (Table S5) were identified between the RM and AA group respectively, and the number of hypermethylated DMPs in the RM group was higher than that of the hypomethylated DMPs in all sample types (Figs. 2 and 3A).

Figure 1 Global DNA methylation across different sample types.

(A) Density plots of DNA methylation of all CpG sites across chorionic villi (green), decidua tissues (orange), and sperm samples (blue). Each line represents one sample. (B) Cluster dendrogram showing the distance relationships among samples of each group.

Figure 2 Volcano plots (A, C, E) and heatmaps (B, D, F) of differentially methylated probes (DMPs) across chorionic villi (A–B), decidua tissues (C–D), and sperm samples (E–F).

The x-axis of the volcano plot is Δ β, and the y-axis is -log10(p value). Red represents hyper-methylated probes, and blue represents hypo-methylated probes.

Figure 3 Distribution of DMPs on the genome.

(A) Statistics of the number of DMPs across different sample types. (B) Distribution of DMPs on the genome.

Distribution of RM-related DMPs and gene enrichment analysis

Among the various sample types, the number of RM-related DMPs in sperm was the least, while that in chorionic villi tissues was the largest (Fig. 3A). Considering that the methylation in promoter, enhancer, intergenic, and DHS regions affects the expression levels of genes, we focused on the DMPs in these regions. The results showed that in all three sample types, the above-mentioned DMPs were mainly distributed in regions such as Body-opensea, IGR-opensea, TSS1500-shore, TSS1500-opensea, and 5′UTR-opensea (Fig. 3B). Among them, the average proportions of Body-island, TSS1500-shore, and IGR-shelf were higher in sperm samples (Fig. 3B). Targeted interrogation of DMPs in intergenic ICRs identified five significant DMPs in RM tissues (Table S6).

Through Gene Ontology (GO) and Kyoto Encyclopedia of Genes and Genomes (KEGG) enrichment analyses of genes related to DMPs, we found that the genes related to DMPs in chorionic villi and decidua tissues were mainly associated with biological processes such as the development and morphogenesis of various tissues and organs (Figs. 4A, 4C), and participated in important pathways such as the PI3K-Akt signaling pathway, Rap1 signaling pathway, Hippo signaling pathway, TGF-beta signaling pathway, and Wnt signaling pathway (Figs. 4B, 4D). In contrast, the genes related to DMPs in sperm were associated with the ionotropic glutamate receptor signaling pathway (Fig. 4E), and participated in pathways such as the phosphatidylinositol signaling system, phospholipase D signaling pathway, calcium signaling pathway, glycerolipid metabolism, and glycerophospholipid metabolism (Fig. 4F). The results of differential DNA methylation indicated that the abnormal epigenetic characteristics in chorionic villi and decidua tissues are involved in the embryonic development process during pregnancy in patients with RM.

Figure 4 Functional enrichment analysis of DMPs related genes.

GO biological processes (A, C, E) and KEGG (B, D, F) enrichment analyses of genes related to DMPs in chorionic villi (A–B), decidua tissues (C–D), and sperm samples (E–F). GO biological processes were defined as significantly enriched when the Holm-Bonferroni corrected P < 0.05, and KEGG pathways were defined as significantly enriched when P < 0.05.

Comparison of DMPs across sample types

The distribution of DMPs in different sample types on chromosomes was relatively scattered (Fig. 5A). However, due to the relatively small number of DMPs in sperm, there was only 1 DMP shared with chorionic villi and decidua tissues, while there were 222 DMPs shared between chorionic villi and decidua tissues (Fig. 5B). Enrichment analysis of the genes related to these DMPs revealed that these genes were mainly associated with biological processes such as the regulation of actin filament organization, atrioventricular valve morphogenesis, regulation of interleukin-2 production, and regulation of monoatomic ion transmembrane transport (Fig. 5C), and participated in pathways such as the Relaxin-signaling pathway, Rap1 signaling pathway, Hippo signaling pathway, and TGF-beta signaling pathway (Fig. 5D).

Figure 5 DMPs across sample types.

(A) Localization of DMPs in chromosomes. The red, blue, and green vertical lines represent DMPs in chorionic villi, decidua, and sperm samples respectively. (B) Venn diagram of DMPs in the three sample types: chorionic villi, decidua, and sperm. (C–D) Network diagrams of GO (C) and KEGG (D) enrichment analyses of genes related to DMPs in chorionic villi and decidua tissues. (E–F) Network diagrams of GO (E) and KEGG (F) enrichment analyses of genes related to common DMPs in chorionic villi and sperm samples.

There were 10 DMPs shared between chorionic villi and sperm samples (Fig. 5B), including DNAH1 (Dynein Axonemal Heavy Chain 1), CAPN1 (Calpain 1), TECPR2 (Tectonin Beta-Propeller Repeat-Containing Protein 2), DGKZ (Diacylglycerol Kinase Zeta), and ULK4 (Unc-51 Like Kinase 4). Enrichment analysis of the genes related to these DMPs revealed that these genes were mainly associated with biological processes such as sperm axoneme assembly, axonemal dynein complex assembly, motile cilium assembly, face morphogenesis, head morphogenesis, body morphogenesis, negative regulation of T cell receptor signaling pathway, regulation of antigen receptor-mediated signaling pathway, and positive regulation of Hippo signaling (Fig. 5E), and participated in pathways such as the pathways of neurodegeneration—multiple diseases pathway, glycerophospholipid metabolism, and apoptosis pathways (Fig. 5F).

Epigenetic differences in imprinted genes related to developmental regulation/immune modulation

Since the abnormal expression of imprinted genes is closely related to placental dysfunction, we focused on the imprinted genes related to DMPs (Table 2). Among them, DMPs were detected in the promoter, enhancer, intergenic, or DHS regions of four imprinted genes, namely CPA4, PRDM16, FUCA1, and HOXB3 in both chorionic villi and decidua tissues (Table 2). Specifically, CPA4 and PRDM16 exhibited hypomethylation in relation to RM in both chorionic villi and decidua tissues (Table 2), and the DMPs of both genes were located in the enhancer regions of the genes. In contrast, the trends of FUCA1 and HOXB3 were completely opposite in these two tissues (Table 2). Additionally, the imprinted gene related to DMPs in sperm was ZNF229 (Table 2).

Table 2 Statistics of imprinted genes related to DMPs across different sample types.

Gene	CHR	Status	Δβ in chorionic villi	Δβ in decidua	Δβ in sperm	
CPA4	7	Imprinted	−0.178	−0.111		
PRDM16	1	Predicted	−0.148	−0.123		
FUCA1	1	Predicted	−0.147	0.124		
HOXB3	17	Predicted	0.102	−0.110		
C6orf145	6	Imprinted	−0.165			
LMNA	1	Imprinted	−0.118			
C10orf91	10	Predicted	−0.101			
CCDC85A	2	Predicted	0.101			
LIN28B	6	Imprinted	0.102			
NKAIN3	8	Predicted	0.108			
GLI3	7	Imprinted	0.111			
APBA1	9	Predicted	0.120			
NAT15	16	Imprinted	0.121			
ERAP2	5	Imprinted	0.122			
GLIS3	9	Imprinted	0.128			
HIST3H2BB	1	Predicted	0.129			
MEST	7	Imprinted	0.141			
PURG	8	Predicted	0.153			
CTNNA3	10	Provisional Data		−0.141		
CDH18	5	Predicted		−0.131		
PLAGL1	6	Imprinted		−0.124		
CHD2	15	Imprinted		−0.114		
MIR296	20	Imprinted		−0.107		
HECW1	7	Imprinted		−0.106		
ATP10A	15	Imprinted		−0.105		
KCNQ1DN	11	Imprinted		0.327		
MAGI2	7	Imprinted		0.105		
RASGRF1	15	Imprinted		0.109		
SLC22A18AS	11	Provisional Data		0.110		
ZFAT	8	Imprinted		0.112		
ZNF229	19	Predicted			0.151	

Experimental validation of methylation effects on imprinted /non-imprintedgenes

To further validate methylation-transcript relationships, qPCR was performed on chorionic villi RNA for selected genes. The imprinted gene LMNA and non-imprinted genes CD59 and IL1RAP exhibited significantly increased expression (P < 0.05) (Figs. 6A–6C), consistent with their hypomethylation status in RM samples. Some of the genes with low transcript abundance (CT > 31) were excluded from analysis due to unreliable quantification. For these, protein-level validation by Western blotting (WB) confirmed elevated expression of both CPA4 (Figs. 6D–6E) and PRDM16 (Figs. 6F–6G) in the RM group of chorionic villi than those in the AA group, inversely correlating with their enhancer hypomethylation (Table 2), indicating that the hypomethylation in the enhancer regions of these genes led to an increase in their protein expression levels in RM patients.

Figure 6 Transcript and protein -level validation of methylation-regulated genes in RM chorionic villi.

(A–C) qPCR analysis of (A) LMNA, (B) CD59 , and (C) IL1RAP expression normalized to β -actin (mean ± SEM; n = 3 biological replicates; *p < 0.05 t-test) (D) Western blotting analysis of CPA4 level in chorionic villi, with β-actin as a control. (E) Boxplot showing the quantification result of the western blotting in (D). (F) Western blotting analysis of PRDM16 level in chorionic villi, with β-actin as a control. (G) Boxplot showing the quantification result of the western blotting in (F). *p < 0.05 t-test .

Discussion

The present study elucidates distinct DNA methylation profiles in gametes and fetal-maternal tissues, identifying potential epigenetic drivers of RM. Among the key findings are the aberrant methylation patterns in imprinted genes, particularly CPA4 and PRDM16, which are differentially methylated in the chorionic villi and decidua. These results suggest that dysregulation of imprinted gene expression due to altered enhancer methylation may contributes to RM by affecting embryo differentiation and immune modulation.

Notably, DMPs in chorionic villi and decidua showed significant enrichment in pathways that were critical for cellular signaling and development, including the PI3K-Akt, TGF-β, and Wnt pathways. This is consistent with results from a Whole-Genome Bisulfite Sequencing (WGBS) study, further highlighting the importance of these pathways in pregnancy-related physiological processes (Irani et al., 2023). The role of the PI3K-Akt signaling pathway in embryonic development and immune regulation has been confirmed by numerous studies (Lin, Wang & Zheng, 2024). This signaling pathway is active in early human embryos and plays a crucial role in maintaining the self-renewal and pluripotency of embryonic stem cells (Wamaitha et al., 2020). The abnormal activation of this pathway is also involved in regulating glucose metabolism in the endometrium, thereby influencing the successful implantation of embryos (Doma Sherpa et al., 2024). Importantly, the maternal epigenetic dysregulation in PI3K-Akt-mediated glucose metabolism complements the fetal proteomic landscape of TCA cycle disruption and ferroptosis in chorionic villi, suggesting a pathogenic synergy between endometrial metabolic insufficiency and placental oxidative stress in recurrent pregnancy loss (Davalieva et al., 2024). The PI3K-Akt signaling pathway is also related to the functional regulation of Th17 cells, which play an important role in the immune balance at the maternal-fetal interface (Chang et al., 2020). The TGF-β signaling pathway plays a key role in the proliferation and development of embryonic cells (Ghimire et al., 2015; Zinski, Tajer & Mullins, 2018). The expression level of TGF-β1 in the placental villi tissues of patients with RM is significantly decreased (Wang et al., 2021), and the reduction in TGF-β1 expression is also associated with recurrent implantation failure, indicating the importance of the TGF-β signaling pathway in endometrial receptivity (Guo et al., 2018). In addition, the Wnt pathway also regulates the growth and development of embryonic stem cells (Nusse & Clevers, 2017). Future studies should further explore the specific mechanisms of these signaling pathways in RM, with the expectation of providing new targets for clinical intervention.

Shared DMP-associated genes across chorionic villi and decidua, such as CPA4 and PRDM16, exhibited hypomethylation in enhancer regions and were associated with an increased protein level in RM patients. CPA4 is overexpressed in multiple types of cancers, and its expression is related to the activities of the Wnt signaling pathway, the MAPK signaling pathway, and the PI3K/AKT/mTOR signaling pathway (Lei et al., 2022). Therefore, CPA4 may influence multiple aspects of embryonic development through these signaling pathways. PRDM16 is a transcription factor with histone methyltransferase activity that regulates cell migration, proliferation, and differentiation (Shull et al., 2020; Van Wauwe et al., 2024). The elevated protein levels of CPA4 and PRDM16 in the villi of RM patients support the hypothesis that dysregulated methylation of imprinted genes contributes to the regulation of gene expression associated with placental dysfunction. Critically, PRDM16 is computationally predicted as parental imprinted gene but only showed significant hypomethylation in both fetal (villi) and maternal (decidua) tissues (Tables S3–S5), implying post-fertilization epigenetic dysregulation (e.g., oxidative/metabolic stress) rather than inherited imprinting errors. CD59 regulates complement-mediated immune responses, while IL1RAP modulates inflammatory and mitogenic signaling pathways (Bai et al., 2015; Zhang et al., 2022). Their dysregulation may disrupt immune tolerance, contributing to RM pathogenesis alongside imprinted genes. Furthermore, the inverse correlation between DNA methylation and protein expression in CPA4 and PRDM16 as well as RNA expression in CD59 and IL1RAP highlights the functional relevance of our methylation findings and suggests that enhancer hypomethylation could serve as a biomarker or therapeutic target in RM.

Notably, the 10 DMPs shared between sperm and chorionic villi (e.g., DNAH1) illuminate paternal epigenetic contributions to RM pathogenesis. DNAH1 is critical for sperm flagellar integrity and may influence fertilization competence when dysregulated in sperm, while its persistent methylation alteration in villi could disrupt embryonic cilia-dependent signaling (Amiri-Yekta et al., 2016; Wang et al., 2017). ULK4 and CAPN1 have been implicated in neurogenesis and apoptotic cell death pathways associated with neurodegenerative disorders (Lang et al., 2016; Metwally et al., 2023), processes critical for early embryonic development. These shared DMPs suggest that paternal epigenetic aberrations may persist post-fertilization and influence placental development, thereby increasing miscarriage risk. Although limited in number, these overlapping DMPs suggest: (i) possible transmission of sperm-derived epigenetic errors to embryonic tissues, or (ii) shared susceptibility loci dysregulated by systemic factors in both gametes and conceptus. This supports the paradigm that paternal epigenetic dysregulation extends beyond sperm quality defects to directly modulate fetal development. Our findings underscore the importance of integrating paternal epigenetic data to fully understand RM etiology.

The results showed that sperm exhibited fewer DMPs, suggesting a relatively fewer paternal contribution to chorionic villi methylation changes compared to maternal contributions. However, this interpretation requires caution due to potential technical and biological constraints: (1) its design covers <2% of CpGs (Sandoval et al., 2011), (2) reduced sensitivity for low-magnitude shifts (<10% Δβ), and (3) inability to capture dynamic methylation reprogramming during spermatogenesis (Siebert-Kuss et al., 2024), which could obscure biologically relevant sperm-derived signals. While these factors may underestimate paternal contributions, our findings consistently highlight maternal-fetal epigenetic coordination as the dominant axis in RM, though future whole-genome bisulfite sequencing will definitively resolve paternal roles.

Our study highlights the importance of integrating multi-tissue methylation data to gain a comprehensive understanding of RM etiology. By focusing on both fetal and maternal tissues, we have identified molecular pathways that bridge these compartments and potentially drive adverse pregnancy outcomes. These insights pave the way for further research into epigenetic therapies aimed at restoring normal methylation patterns in affected pregnancies. However, several limitations must be acknowledged. (1) The sample size is relatively small due to the challenge of obtaining high-quality paired tissues from RM and control cases. This limitation reduces the generalizability of our findings and underscores the need for larger cohorts in future studies. (2) The limited detection of ICR-associated DMPs (n = 5) and absence of sperm methylation changes for imprinted genes likely reflect inherent constraints of the 450K array, which covers <2% of CpGs and under-samples conserved intergenic regions (Sandoval et al., 2011), while its promoter-centric bias underrepresents non-coding elements, limiting insight into how methylation effects differ by genomic context (Jones, 2012). This probe bias toward gene promoters inherently underrepresents non-coding regulatory elements, restricting functional interpretation of methylation changes in context-specific genomic compartments. Notably, the limited number of detected significant imprinted genes in RM sperm contrasts with targeted pyrosequencing studies reporting multiple aberrant loci (Khambata et al., 2023; Khambata et al., 2021)—a discrepancy likely attributable to the inherent technical limitations of the 450K array platform, as detailed above. Future studies employing whole-genome bisulfite sequencing will provide comprehensive profiling of ICR-specific methylation dynamics in sperm. (3) Attempts to validate CPA4 and PRDM16 via qPCR were hindered by low transcript abundance (CT >  31), likely reflecting tissue-specific expression patterns. Future studies using sensitive methods (e.g., digital PCR) or alternative tissues (e.g., trophoblast primary cells) may overcome this limitation. (4) While we identified significant DMPs and their associated pathways, the causal relationship between methylation changes and RM remains uncertain. Functional studies, including cell-based assays and animal models, are essential to validate the biological roles of candidate genes. For example, knockdown or overexpression experiments in trophoblast cell lines could elucidate the specific contributions of these genes to placental development and immune regulation. Similarly, animal models with tissue-specific alterations in gene methylation could provide insights into their role in embryonic development and RM pathology.

Conclusions

This study provides a comprehensive analysis of DNA methylation patterns in gametes and fetal-maternal tissues associated with recurrent miscarriage (RM), revealing significant epigenetic dysregulation that may contribute to pregnancy failure. Our findings highlight the critical role of imprinted genes, particularly CPA4 and PRDM16, in the pathogenesis of RM. These genes exhibited distinct hypomethylation in enhancer regions, corresponding to elevated transcript/protein expression levels in RM villi tissues, suggesting that dysregulated methylation of imprinted genes may disrupt normal embryonic development and maternal-fetal immune interactions. The identification of differentially methylated probes (DMPs) and their enrichment in critical pathways such as the PI3K-Akt, TGF-beta, and Wnt signaling pathways underscores the potential mechanisms underlying RM. These pathways are essential for tissue morphogenesis, immune regulation, and cell signaling, indicating that epigenetic alterations in these regions may significantly impact pregnancy outcomes.

In conclusion, our findings emphasize the critical role of epigenetic regulation, particularly in imprinted genes, in RM pathogenesis. By identifying key DMPs and validating their functional impact, this study provides novel insights into the molecular basis of RM. Future studies should expand on these findings by investigating larger cohorts and exploring the potential for targeted interventions to mitigate epigenetic dysregulation in RM patients.

Supplemental Information

Supplemental Information 1 Representative phase-contrast micrograph of purified sperm after swim-up separation

Motile sperm isolated from semen samples show characteristic morphology with intact flagella (Scale bar: 20 μ m).

Supplemental Information 2 Sequences of primers used for RT-qPCR

Supplemental Information 3 The datasets included in this study

Supplemental Information 4 The differentially methylated probes (DMPs) identified in chorionic villi between recurrent miscarriage (RM) and artificial accidental termination of pregnancy (AA) groups

Supplemental Information 5 The differentially methylated probes (DMPs) identified in decidua between recurrent miscarriage (RM) and artificial accidental termination of pregnancy (AA) groups

Supplemental Information 6 The differentially methylated probes (DMPs) identified in sperm between recurrent miscarriage (RM) and artificial accidental termination of pregnancy (AA) groups

Supplemental Information 7 The DMP in intergenic regions harboring imprinting control regions (ICRs) between recurrent miscarriage (RM) and artificial accidental termination of pregnancy (AA) groups

Supplemental Information 8 Photos of the full-length blots for western blot experiments

Additional Information and Declarations

Competing Interests

Author Contributions

Human Ethics

Microarray Data Deposition

Data Availability

The authors declare there are no competing interests.

Yanru Niu performed the experiments, analyzed the data, prepared figures and/or tables, authored or reviewed drafts of the article, and approved the final draft.

Lanlan Yin performed the experiments, analyzed the data, prepared figures and/or tables, authored or reviewed drafts of the article, and approved the final draft.

Yulan Zhou performed the experiments, prepared figures and/or tables, and approved the final draft.

Xiaoyan Pang performed the experiments, prepared figures and/or tables, and approved the final draft.

Yunqinq Li performed the experiments, prepared figures and/or tables, and approved the final draft.

Cailing Peng analyzed the data, prepared figures and/or tables, and approved the final draft.

Meihua Yao analyzed the data, prepared figures and/or tables, and approved the final draft.

Guoling Zhang analyzed the data, prepared figures and/or tables, and approved the final draft.

Kaijie Yang analyzed the data, prepared figures and/or tables, and approved the final draft.

Tianzhong Ma conceived and designed the experiments, authored or reviewed drafts of the article, and approved the final draft.

The following information was supplied relating to ethical approvals (i.e., approving body and any reference numbers):

This study was approved by the Ethics Committee of the Affiliated Hospital of Guangdong Medical College (YJY2018084), and all subjects signed the informed consent forms.

The following information was supplied regarding the deposition of microarray data:

The DNA methylation datasets generated and analyzed during this study are available at NCBI Gene Expression Omnibus (GEO): GSE287809.

The following information was supplied regarding data availability:

The DNA methylation datasets generated and analyzed during this study are available at NCBI Gene Expression Omnibus (GEO): GSE287809.

The quantification result of the western blotting and the code used for data processing and analysis are available at Zenodo: Ma, T. (2025). Comprehensive analysis of DNA methylation patterns in recurrent miscarriage: imprinted genes and their regulation across sperm and fetal-maternal tissues. Zenodo. https://doi.org/10.5281/zenodo.14860255.

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
