# Peer review of "Comprehensive analysis of DNA methylation patterns in recurrent miscarriage: imprinted/non-imprinted genes and their regulation across sperm and fetal-maternal tissues"

_PeerJ, doi:10.7717/peerj.20125_

## Round 0.1 · original submission · Major Revisions

Two reviewers have provided detailed comments on your manuscript.

Reviewer 1 ·

Basic reporting

1. Relevant prior literature not mentioned in the Introduction
2. Differential Methylome data with Beta values for the samples analyzed by the authors should be included

Experimental design

None

Validity of the findings

1. Studies are available in the literature on methylome studies in the different tissues, i.e., chorionic villi, decidua, and spermatozoa. The present study has analyzed all 3 together, analyzing samples collected by them as well as from publicly available datasets

Additional comments

Introduction:
1. Some of the published papers on sperm DNA methylation in RM couples related to imprinted genes and the genomewide methylome have not been cited
2. Line 58-59: “Studies have demonstrated that abnormal DNA methylation is closely associated with RM (Demond et al. 2019)”. This paper reports on recurrent Hydatiform mole and not RM

Materials and Methods:
1. The authors should mention in M & M about the use of publicly available 2 datasets on decidua and chorionic villi used. A brief description of those 2 studies should be mentioned as to whether gestational age matched with their samples
2. Sperm DNA extraction: any specific treatment given is not mentioned. Was it ensured that DNA bound to protamines was extracted? In addition, how was somatic cell contamination in the spermatozoa samples ruled out?

Results:
1. The differentials obtained in the sperm, chorionic villi, and decidua between control and RM with their FDR and Beta values should be given as supplementary data
2. Common differentials between spermatozoa and chorionic villi should be analyzed, which may bring out the role of paternal epigenetic factors
3. Only validated 2 genes at protein levels, more genes, both imprinted as well non-imprinted, should be validated at transcript level and correlated with methylation status
4. For imprinted genes, intergenic regions are important as some imprinted gene clusters have their imprint control region in these regions. DMCs in this region should also be analyzed

Discussion:
1. Lines 239-241: "These results suggest that dysregulation of imprinted gene expression due to altered enhancer methylation contributes to RM by affecting embryo differentiation and immune modulation". The inference is based on the result of 2 genes, then what is the significance of the other differentials? Whether altered methylation contributes to RM or is the result of RM cannot be inferred from the result obtained
2. The authors need to discuss how an increase in the expression of the 2 imprinted genes can lead to RM
3. Lines 277-280: "Our study highlights the importance of integrating multi-tissue methylation data to gain a comprehensive understanding of RM etiology. By focusing on both fetal and maternal tissues, we have identified molecular pathways that bridge these compartments and potentially drive adverse pregnancy outcomes". This point needs to be elaborated on as to how the results from the study have helped to bridge the maternal and fetal compartments to understand the adverse pregnancy outcome. In fact, getting common differentials between spermatozoa and chorionic villi would be able to infer the significance of paternal factors in early embryo development

Reviewer 2 ·

Basic reporting

• The manuscript presents a comparative DNA methylome analysis across paternal sperm, maternal decidua, and fetal chorionic villi.
• Rationale and methodology are aligned with the study hypothesis.
• Sufficient literature provided.
• Several improvements and clarifications are needed for publication.

Experimental design

1. The manuscript is original and within the aims and scope of the journal.
2. The research question is well defined.
3. Rigorous investigation performed to a high technical and ethical standard.
4. Methodology Details: Include a brief description of the Infinium Human Methylation 450 BeadChip protocol.
5. Semen Parameters: Were semen parameters assessed for male participants? If yes, include this data.

Validity of the findings

Comments and suggestions:
1. Definition Consistency
of RM is defined as "3 or more" in the abstract and "2 or more" in the introduction. This should be consistent throughout.

2. Figure Labeling Consistency
o In heatmaps, group designations (AA vs RM) are reversed across tissues, specifically in the sperm DMPs heatmap. Standardize group order in figures to avoid confusion.

3. Comparison with Published Datasets
o Consider comparing findings with other published datasets:
 Example: Irani et al. (2023) sperm methylation in RSA.
 Davalieva et al. (2024) chorionic villi proteomics in RM.
o Such comparisons could strengthen and validate findings.
o Correlation analysis can be performed.
o The authors should provide the complete list of differentially methylated positions (DMPs) identified in all three tissues (sperm, chorionic villi, and decidua) as supplementary Excel files.

4. Prdm16 Methylation Analysis
o If Prdm16 is predicted as imprinted, analyze its methylation status in sperm to evaluate parent-of-origin expression.

5. No Methylation Changes in Sperm for Imprinted Genes
o Discuss the possible biological implications of the absence of changes in sperm for imprinted genes.

6. Beyond Imprinted Genes
o Expand analysis beyond imprinted genes:
 Compare DMPs between sperm vs chorionic villi.
 Compare DMPs between decidua vs chorionic villi.
o This could provide insight into maternal vs paternal contributions.

7. Paternal Contribution Assessment
• Based on data, is it valid to conclude that the father contributes less to DNA methylation changes in chorionic villi than the mother?

8. Post-fertilization Reprogramming
• Address post-fertilization methylation reprogramming and how some loci escape reprogramming. Relate this to your findings.

9. CpG Site Specificity in Imprinted Genes
• Address the possibility that total methylation may obscure CpG site-specific changes.
• Focus on DMRs and imprinting control regions for detailed analysis.

10. Limitations of 450K Array
• Acknowledge the array’s inability to capture CpG site-specific changes.
• Mention that methylation in promoters vs introns has different functional outcomes.

11. Sex of Chorionic Villi Samples
• Was fetal sex determined? If not, consider analyzing it, as sex-specific effects may influence results.

12. Comparison with Khambata et al. (2020)
• Only one imprinted gene shows significance in your study vs several in Khambata et al. 2020. Discuss this discrepancy.

13. Literature Discrepancies
• Address inconsistencies across studies regarding imprinted gene methylation in sperm from RM cases.
• Suggest possible causes for these differences.

14. Global Methylation Assessment
• Has global DNA methylation been assessed in all three tissues? If not, suggest including this analysis.

Additional comments

No comments.

---

## Round 0.2 · Minor Revisions

Dear Dr. Ma,

Both reviewers acknowledged that the authors have addressed most of the previous concerns and significantly improved the manuscript through additional data analysis and manuscript revisions. However, several minor but important revisions remain to be addressed:

Reviewer 1:
DNA extraction protocol: Authors must clarify and provide a proper reference for the DNA extraction method used for spermatozoa, particularly regarding disulfide bond disruption.

Conclusion update: The conclusion must be updated to reflect the findings related to non-imprinted genes and their association with recurrent miscarriage (RM).

Paternal contribution: The potential role of the 10 differentially methylated genes found in both sperm and chorionic villi in RM should be explicitly discussed.

Title revision: The manuscript title should be updated to include reference to non-imprinted genes, reflecting the expanded scope of the findings.

Reviewer 2:
Definition consistency: The definition of recurrent miscarriage (RM) must be made consistent throughout the manuscript. The authors have stated the correction but failed to implement it in the abstract and introduction.

Reviewer 1 ·

Basic reporting

The manuscript has been revised considerably as per suggestions, doing additional analysis of the data.
There are a few points that need to be addressed, which are mentioned in the appropriate section.

Experimental design

All suggestions incorporated. One clarification is needed, as mentioned below

1. DNA extraction from spermatozoa:
The authors should give a reference to the protocol followed. Usually, sperm DNA extraction requires treatment with a detergent like SDS as well as dithiothreitol or 2 2-mercaptoethanol to break the bisulfide bonds in protamines, in addition to Proteinase K treatment. Please clarify.

Validity of the findings

1. As per the result of the revised analysis of the data, aberrant methylation of both imprinted and non-imprinted genes is associated with RM. Hence, the conclusion needs to be extended to incorporate the other genes validated.

2. In addition, based on the 10 genes identified between sperm and chorionic villi, their potential involvement in paternal contribution to RM should also be mentioned.

3. Based on above mentioned points, the title needs to be modified to incorporate non-imprinted genes

Additional comments

-

Reviewer 2 ·

Basic reporting

The authors have responded to all my queries and made appropriate changes to the manuscript, except for one query:

1. Definition Consistency
of RM is defined as "3 or more" in the abstract and "2 or more" in the introduction. This should be consistent throughout.

Response:
We sincerely thank the reviewer for identifying this inconsistency in the definition of recurrent miscarriage (RM). To ensure clarity and alignment with current clinical guidelines, we have revised the manuscript to standardize the definition of RM as "two or more " throughout the text.

These changes are not made in the abstract and introduction sections.

Experimental design

-

Validity of the findings

The authors have performed additional analysis and elaborated on the discussion section.

---

## Round 0.3 · accepted · Accept

Congratulations on the acceptance of your manuscript.

Reviewer 1 ·

Basic reporting

no comment

Experimental design

no comment

Validity of the findings

No comment

Additional comments

The comments addressed and the MS accordingly revised
MS can be accepted

Reviewer 2 ·

Basic reporting

The authors have revised the manuscript in response to my comments and suggestions.

Experimental design

The authors have revised the experimental design in response to my comments and suggestions.

Validity of the findings

The authors have validated the findings in response to my comments and suggestions.

Additional comments

No comments